# Cell Fate following Irradiation of MDA-MB-231 and MCF-7 Breast Cancer Cells Pre-Exposed to the Tetrahydroisoquinoline Sulfamate Microtubule Disruptor STX3451

**DOI:** 10.3390/molecules27123819

**Published:** 2022-06-14

**Authors:** Scott D. Hargrave, Anna M. Joubert, Barry V. L. Potter, Wolfgang Dohle, Sumari Marais, Anne E. Mercier

**Affiliations:** 1Department of Physiology, Faculty of Health Sciences, University of Pretoria, Pretoria 0001, South Africa; shargrave09@gmail.com (S.D.H.); annie.joubert@up.ac.za (A.M.J.); sumari.marais@up.ac.za (S.M.); 2Medicinal Chemistry & Drug Discovery, Department of Pharmacology, University of Oxford, Mansfield Road, Oxford OX1 3QT, UK; barry.potter@pharm.ox.ac.uk (B.V.L.P.); wolfgang.dohle@pharm.ox.ac.uk (W.D.)

**Keywords:** cancer, 2-methoxyestradiol, tetrahydroisoquinoline analog (STX3451), radiosensitization, reactive oxygen species, apoptosis, micronuclei, DNA damage response (DDR), microtubule disrupting agent (MDA), radiation

## Abstract

A tetrahydroisoquinoline (THIQ) core is able to mimic the A and B rings of 2-methoxyestradiol (2ME2), an endogenous estrogen metabolite that demonstrates promising anticancer properties primarily by disrupting microtubule dynamic instability parameters, but has very poor pharmaceutical properties that can be improved by sulfamoylation. The non-steroidal THIQ-based microtubule disruptor 2-(3-bromo-4,5-dimethoxybenzyl)-7-methoxy-6-sulfamoyloxy-1,2,3,4-tetrahydroisoquinoline (STX3451), with enhanced pharmacokinetic and pharmacodynamic profiles, was explored for the first time in radiation biology. We investigated whether 24 h pre-treatment with STX3451 could pre-sensitize MCF-7 and MDA-MB-231 breast cancer cells to radiation. This regimen showed a clear increase in cytotoxicity compared to the individual modalities, results that were contiguous in spectrophotometric analysis, flow cytometric quantification of apoptosis induction, clonogenic studies and microscopy techniques. Drug pre-treatment increased radiation-induced DNA damage, with statistically more double-strand (ds) DNA breaks demonstrated. The latter could be due to the induction of a radiation-sensitive metaphase block or the increased levels of reactive oxygen species, both evident after compound exposure. STX3451 pre-exposure may also delay DNA repair mechanisms, as the DNA damage response element ataxia telangiectasia mutated (ATM) was depressed. These *in vitro* findings may translate into *in vivo* models, with the ultimate aim of reducing both radiation and drug doses for maximal clinical effect with minimal adverse effects.

## 1. Introduction

Microtubule disrupting agents (MDAs) remain a mainstay in the toolbox of cancer therapeutics. Their clinical application walks a tightrope between their anticancer effects, the side effects incurred and the development of tumor resistance, whilst managing their pharmacokinetic limitations [1]. 2-Methoxyestradiol (2ME2) is an endogenous estrogen metabolite that has shown promise as an anticancer agent inter alia by disrupting microtubule dynamics via binding to the colchicine binding site [2]. The compound demonstrated cytotoxicity to a wide range of neoplastic cells *in vitro* at micromolar concentrations by inducing apoptosis via the intrinsic and extrinsic pathways [3,4]. Additionally, 2ME2 functions independently of hormone receptors as well as preferentially sparing non-neoplastic cell counterparts, which translates into better tolerated side-effect profiles *in vivo* [5]. The compound’s inability to function as a substrate of multi-drug efflux pumps is an added advantage for preventing and/or circumventing chemotherapeutic resistance [6]. 2ME2 also demonstrated antiangiogenic properties, a property useful in preventing neovascularization of expanding tumors [7]. However, clinical trials were halted after phase II due to the compound’s low bioavailability and rapid metabolism [8].

In an attempt to improve the pharmacokinetics and enhance the cytotoxicity, a variety of 2ME2 analogs were designed and synthesized by various research groups [9,10,11]. Modifications at the C3 and C17 positions of the steroid structure allowed escape from 17β-hydroxysteroid dehydrogenase metabolism, while sulfamoylation at C3 also improved not only oral bioavailability, but also the anti-mitotic and spindle disruption capacities, as seen with 2-methoxyestradiol-3,17-*O*,*O*-bis-sulfamate (STX140) [12]. STX140 displayed a 10-fold increase in anti-mitotic capacity and a 60-fold increase in antiangiogenic effect when compared to the parent compound, with IC_50_ values at nanomolar concentrations [13]. This analog was less toxic to non-cancerous cells *in vitro* [14,15]. In addition, the analog displayed better oral bioavailability, although myelosuppression and tolerated steroidal excipients remained problematic [1]. Progress has been reviewed [16,17].

Non-steroidal microtubule disruptors were designed to provide more structural diversity and improve the clinical shortcomings of 2ME2 and the potency of sulfamoylated derivatives such as STX140. A structure–activity relationship analysis based on 2ME2 and STX140 led to the design and optimization of tetrahydroisoquinoline (THIQ)-based steroidomimetic and chimeric microtubule disruptors [18,19,20]. Structural elements such as the 2-methoxy group of 2ME2 and the 3-sulfamoyloxy group of STX140 were combined in the 7-methoxy-6-sulfamoyloxy THIQ core in order to mimic the steroidal A,B ring system. The steroidal D ring was mimicked by attaching various functionalized benzyl groups at the N2 position of the THIQ core. The chosen benzyl groups usually had one or several hydrogen-bond acceptors attached in order to mimic the 17β-position of the parent steroidal compounds.

One of these THIQ-based compounds, 2-(3-bromo-4,5-dimethoxybenzyl)-7-methoxy-6-sulfamoyloxy-1,2,3,4-tetrahydroisoquinoline (STX3451) (Figure 1), has been shown to induce a G_2_–M phase cell cycle arrest, apoptosis and autophagy, whilst demonstrating a favorable toxicity profile *in vivo* (Appendix A) [21].

Radiotherapy and chemotherapeutics are mainstays of conventional breast cancer treatment. Both of these therapies are accompanied by undesirable side effects and therapeutic failure. Steel and Peckham presented categories by which combined chemotherapy–radiotherapy could enhance clinical effectivity, which included special cooperation, improved tumor radiosensitivity and toxicity independence [22]. The first concept encompassed the use of radiation for local tumor control with chemotherapy for control of distant metastasis. Enhanced tumor response relies on the chemotherapeutic agent having a direct effect on the neoplastic cells to make them more sensitive to the radiation. Lastly, toxicity independence speaks to the concept that both the drug and the radiation work synergistically to increase tumor cytotoxicity whilst sparing normal tissue. Over the years, discovering agents that are able to change the intracellular response to radiation to enhance tumor susceptibility has become an important aim. These targeted agents include those that disrupt the DNA damage response, apoptosis pathways, transcription factors, growth factor receptors and cytoplasmic signal transduction [23].

Investigation of chemotherapeutic agents that augment radiation damage to targeted cells whilst limiting unwanted effects on the surrounding tissue remains a goal on the research agenda [24]. Agents that have garnered interest include agents altering tumor hypoxia, MDAs and caspase activators, amongst others. 2ME2 and its sulfamoylated analogs address those parameters and indeed have been shown to pre-sensitize prostate and breast cancer cells to radiation whilst selectively sparing non-neoplastic cells [25,26,27,28]. The non-functional microtubules result in a G_1_ and/or a G_2_–M phase block, which are described as the most radiosensitive phases of the cell cycle [29,30]. Additionally, increased generation of reactive oxygen species (ROS) from exposure to the 2ME2/2ME2 analogs may contribute to those formed by the intracellular water–photon reaction from the radiation, resulting in intrinsic apoptosis and DNA damage. This DNA damage results in the majority of radiation-induced cell deaths [31,32]. Defective shuttling due to microtubule abrogation may cause disrupted autophagy and impaired shuttling of DNA repair proteins [21,29,33]

As yet, there has been no investigation of the non-steroidal MDA STX3451 as a radio-sensitizing agent. Building on current knowledge, the aim of this study was to investigate the cellular responses of hormone receptor-positive and receptor-negative breast cancer lines receiving radiation following pre-sensitization with STX3451. To align with the literature, the design aimed to assess the effect of this chemo-radiation on induction of apoptosis and long-term cell survival, cell cycle blockade, the extent of DNA damage and the DNA damage response [34].

## 2. Results

### 2.1. STX3451 Pre-Exposure Significantly Increases the Cytotoxicity of Radiation

In a dose–response study over 48 h STX3451 displayed cytotoxicity at nanomolar concentrations. The concentration of drug at which 50% of growth inhibition occurred (GI_50_) was determined to be 0.074 ± 0.007 μM and 0.065 ± 0.007 μM for MCF-7 and MDA-MB-231 cells, respectively (Figure 2). In previous studies, it had been determined that 6 Gy radiation was sufficient to decrease the cell viability to 80% [27,35]. Thus, in the chemo-radiation group, 6 gray (Gy) radiation was applied to cells already exposed to the drug concentration range for 24 h. Cells were allowed another 24 h incubation for a total 48 h exposure time. Both cell lines displayed significantly increased cytotoxicity when exposed to radiation after pre-sensitization, particularly at lower doses of STX3451 in the MCF-7 cells (GI_50_ of 0.036 ± 0.04 μM and 0.037 ± 0.006 μM for MCF-7 and MDA-MB-231 cells, respectively).

### 2.2. STX3451 and Radiation Induce a Metaphase Block, as Both Individual Modalities and in Combination

To investigate the effect of STX3451 treatment on mitotic progression, flow cytometric cell cycle analysis using propidium iodide (PI) was conducted in both cell lines (Figure 3). Cells were exposed to the compound 24 h prior to 6 Gy radiation, and the experiment terminated 24 and 48 h thereafter. Paclitaxel, a microtubule stabilizer, was used as a positive method control as it induces a metaphase block and apoptosis [36]. Cells exposed to STX3451 for the equivalent amount of time and cells exposed to radiation alone constituted the experimental controls. Our results indicate cell cycle arrest in the G_2_–M phase in both cell lines in response to STX3451 exposure. MCF-7 cells displayed 52.59 ± 3.66% and 58.13 ± 2.05% in the G_2_–M phase after 24 h in response to STX3451 and the combination treatment, respectively. After 48 h, MCF-7 cells showed 63.20 ± 0.76% and 64.70 ± 1.61% of cells in G_2_–M in response to STX3451 and combination treatment, respectively. In the 24 h timeline, MDA-MB-231 cells showed 49.44 ± 0.88% and 54.80 ± 4.74% in G_2_–M in response to STX3451 and combination, respectively. The number of MDA-MB-231 cells in G_2_–M increased after 48 h in response to STX3451 (62.61 ± 8.35%), while combination-treated cells showed that 54.70 ± 2.04% of the cells were in metaphase.

MCF-7 cells displayed no difference in the sub-G_1_ phase between the two timeline points, with the number of apoptotic cells increasing in both the drug and combination exposure. There was a decrease in the S phase at 48 h in all treatment conditions for the 24 h values, and the proportion of cells in metaphase increased slightly in the drug and combination conditions at 48 h. Of significance was the accumulation of cells in the G_2_–M phase at both time points, which were similar in the drug, radiation and combination samples. At 48 h, the drug and combination samples had less viable cells compared to the radiation control. A similar pattern (with small deviations) was observed for the MDA-MB-231 cells.

Results thus indicate that the cell cycle block in metaphase may have mechanisms originating from both the drug and the radiation treatments. Measuring this as an outcome, neither synergistic nor additive mechanisms can be demonstrated in the combination samples. Effects of the drug in combination with radiation did appear to affect the viability of the cells and increase apoptosis, although the amount was slight at this dose and time point.

### 2.3. Early Apoptosis Induction Appears to Be Mainly Attributed to the Drug Exposure

To further investigate the sub-G_1_ populations seen in the cell cycle analysis, flow cytometric analysis of apoptosis induction using annexin V–fluorescein isothiocyanate (FITC) was carried out (Figure 4). In the early stages of apoptosis, phosphatidylserine in the cell membrane is externalized, to which FITC-conjugated annexin V binds [37]. The cytotoxicity induced by the combination treatment was evident from the significant decrease in viable cells (39.68 ± 0.86% in MCF-7 and 28.96 ± 0.26% in MDA-MB-231), accompanied by a significant increase in apoptotic cells (MCF-7 (44.64 ± 1.63% in MCF-7 cells and 68.47 ± 3.82% in MDA-MB-231 cells), compared to the vehicle control. This was in contrast to the vehicle controls, in which MCF-7 cells had 91.94 ± 1.23% viable cells and 5.98 ± 2.31% apoptotic cells. MDA-MB-231 cells exposed to DMSO showed 87.20 ± 1.13% viable cells and 9.91 ± 1.13% apoptotic cells. STX3451 seemed to increase the proportion of necrotic cells, in both the drug control and combination samples, particularly in the MCF-7 cells.

As with the cell cycle analysis, apoptosis induction at an early timeframe in the combination treatment was similar to the drug-only control. The radiation samples at this time had only a very small decrease in cell viability, and no to little increase in apoptotic cells. Thereafter, there is a significant increase in metaphase cells and a concurrent increase in apoptotic cells (albeit small at these does). The long-term viability of these treated samples was further investigated with clonogenic studies.

### 2.4. Long-Term Viability Studies Indicated That STX3451 and 6 Gy Radiation Display a Synergistic Effect in Inhibiting Colony Formation

To assess the long-term survival and replicative ability of treated cells, clonogenic assays were conducted (Figure 5). Both cell lines displayed a significant decrease in the number of colonies formed in response to both STX3451 and radiation exposure compared to vehicle control cells. The two treatment modalities appeared to have a synergistic effect on decreasing long-term viability, with a statistically significantly lower number of colonies when compared to the individual treatments.

The surviving fraction of colonies formed by MCF-7 and MDA-MB-231 cells exposed to STX3451 decreased to 0.06 ± 0.02% and 0.04 ± 0.007% of the negative control value, respectively. The surviving fraction of MCF-7 cells exposed to 6 Gy radiation decreased to 0.09 ± 0.01%, while that of MDA-MB-231 cells exposed to 6 Gy radiation decreased to 0.06 ± 0.01% when compared to the negative control. MCF-7 cells exposed to combination treatment showed a significant decrease in surviving fraction to 0.003 ± 0.002% when compared to the negative control. When compared to STX3451- and radiation-exposed cells, combination-treated cells showed significant decreases in surviving fraction to 0.05 ± 0.001% and 0.04 ± 0.001%, respectively. MDA-MB-231 cells exposed to combination treatment showed similar results with surviving fraction decreasing significantly to 0.001 ± 0.001% compared to negative control cells. Using STX3451 and radiation-exposed cells as a baseline, the surviving fraction of combination-treated cells decreased significantly to 0.02 ± 0.001% in comparison to STX3451 and 0.01 ± 0.001% when compared to radiation.

The combination-treated cells displayed a synergistic effect between STX3451 and radiation in decreasing colony formation in both cell lines. Thus, the long-term survival of cells exposed to the investigated treatment modality is significantly decreased. Mercier et al. showed the microtubule-disrupting effects of the 2ME2 sulfamoylated analogs ESE-16 and ESE-15-one to be reversible [29]. If this reversibility is retained in STX3451, this may account for the ability of STX3451-exposed cells to form more colonies relative to combination-treated cells. The hypothesis we generated was that pre-treated cells were blocked in the G_2_–M phase and thus sustained enough DNA damage to inhibit colony formation.

### 2.5. Radiated Cells Pre-Treated with STX3451 Display Cell Rounding, Decreased Cell Density and Evidence of Apoptosis

We compared the morphological response of pre-treated cells to radiation to cells exposed to either the drug or radiation alone using polarization-optical transmitted light differential interference contrast microscopy (PlasDIC). Neither cell line showed differences in morphology between the negative and vehicle control cells. Etoposide was used as a positive control. MCF-7 and MDA-MB-231 cells exposed to STX3451 demonstrated a decreased cell density, as well as cell rounding, most likely indicative of cells in metaphase (Figure 6). There was evidence of cellular distress, with marked cell protrusions evident. Some apoptotic bodies were identifiable. Radiation-exposed samples displayed the rounded cells, most likely in keeping with the G_2_–M block demonstrated in the cell cycle analysis, with cell densities still intact and not much evidence of apoptosis. This is in keeping with the apoptosis studies. Samples treated with radiation after drug sensitization demonstrated a decrease in cell density, evidence of cell rounding and apoptosis. Some cells also appear to have a flattened morphology.

Microscopy confirmed that radiation exposure after STX3451 pre-treatment decreased cell viability and displayed features of a metaphase block, seen by the presence of more rounded cells. Furthermore, evidence of apoptosis tied the morphological finding with the cytometric data. Future studies should investigate the induction of senescence with this treatment modality.

### 2.6. Chemo-Radiation Induces Greater DNA Damage Than the Individual Modalities

To investigate the extent of chromosomal damage in response to the various treatments, the cytokinesis-block micronucleus (Mn) technique was used. Because this technique requires binucleated cells, cells were exposed to 12 mM cytochalasin B 2 or 24 h after exposure to 6 Gy radiation.

Both MCF-7 and MDA-MB-231 cells showed a significantly increased total number of micronuclei at both termination timepoints when exposed to STX3451 or radiation (Figure 7). At 2 h post-radiation, MCF-7 cells exposed to both STX3451 (124.5 ± 50.20 Mn) and radiation (99 ± 25.46 Mn) showed a statistically significant increase in the total number of Mn compared to the DMSO vehicle control samples (15 ± 10.58 Mn) (Figure 7(Bi)). Cells exposed to the experimental combination showed a significantly increased number of Mn (358 ± 68.90 Mn) in comparison to radiation- and STX3451-treated cells. Similarly, at 24 h post-radiation, STX3451-exposed cells (134.5 ± 75.66 Mn) and radiation-exposed cells (198.5 ± 13.44 Mn) displayed significantly increased Mn formation. Combination-treated cells showed a significant increase in the number of Mn formed (322.33 ± 41.04 Mn), more so than in the experimental controls (Figure 7(Biii)).

The total Mn count of MDA-MB-231 cells exposed to STX3451 was 170 ± 28.28 Mn at 2 h and 141.5 ± 23.35 Mn at 24 h, whereas in the radiation-exposed samples, the total Mn count was 92.67 ± 11.37 Mn at 2 h and 173.33 ± 24.54 Mn at 24 h (Figure 7(Bii,Biv)). The chemo-radiated cells had significantly more damage than the experimental controls, with 384 ± 37.17 Mn at 2 h and 347 ± 27.84 Mn at 24 h.

Of interest is the observation that STX3451 caused a significant amount of DNA damage in the absence of radiation. Further investigations into the genotoxic properties of this compound may well be of interest. When STX3451 was combined with radiation, the total amount of DNA damage was significantly higher than that in either individual treatment. Furthermore, a greater degree of DNA damage per cell in was observed in the combination treatment when compared to both DMSO and the individual treatment conditions.

When compared to the STX3451-exposed MCF-7 cells, the combination samples showed a significantly higher percentage of cells containing 2 (43.33 ± 8.74%), 4 (7 ± 4.4%) and 5 (2.67 ± 1.53%) Mn (Figure 7(Ai,Aii). Similarly, when compared to the radiation-expo-sed samples, the combination-treated cells showed significantly increased percentages of cells containing 1 (119.33 ± 13.42%), 2 (43.33 ± 8.74%), 3 (19.67 ± 9.87%), 4 (7 ± 4.36%) and 5 (2.33 ± 1.53%) Mn. Similar trends were observed in the MDA-MB-231 cells. Of interest were the different patterns of Mn formation seen between the radiation-only exposed samples and those that had drug treatment. Without the drug, Mn formation increased significantly between the early (2 h) and later (24 h) timeframe in both cell types in response to 6 Gy radiation. STX3451 exposure was deemed to permit a rapid and sustained increase in Mn formation right from the early termination point.

From these results, it was evident that breast cancer cell exposure to 0.07 µM STX3451 for 24 h prior to radiation not only increased the total number of Mn formed, but also increased the extent of the damage in more cells than in the experimental controls. Furthermore, the drug exposure facilitated a more rapid and sustained pattern of Mn formation as opposed to the radiation controls.

### 2.7. Increase in Reactive Oxygen Species Is Attributed Largely to STX3451 Exposure

Both radiation and exposure to 2ME2 and its analogs increase the production of intracellular reactive oxygen species (ROS), which contribute to genotoxicity as well as induction of the autophagy and the intrinsic apoptotic pathway [27,38] We investigated the effect of STX3451 and radiation, and the combination thereof, on the formation of ROS at both 24 h and 48 h after radiation.

Superoxide detection was significantly increased in the combination-treated MCF-7- and MDA-MB-231 cells 24 and 48 h after radiation, which was more so than the radiation control in the MDA-MB-231 cells at 48 h (Figure 8). Radiation-exposed MCF-7 cells showed a significant increase in ROS formation after 24 h (1.56 ± 0.30-fold increase) which decreased to baseline levels after 48 h (1.52 ± 0.27-fold). MCF-7 cells exposed to combination treatment showed a significant increase in superoxide formation (1.80 ± 0.22-fold increase) compared to the STX3451 experimental control (1.24 ± 0.14-fold increase) 24 h after radiation. Combination-treated cells showed a significant increase in ROS formation compared to the radiation-exposed cells (1.56 ± 0.30-fold increase) 48 h post-radiation. MDA-MB-231 cells exposed to both STX3451 and radiation controls displayed significant fold increases in ROS formation 24 (2.89 ± 0.22-fold increase) and 48 (2.29 ± 0.13-fold increase) h after radiation compared to vehicle control cells. Combination-treated MDA-MB-231 cells also displayed a significant increase at the 48-h timeline compared to radiation-exposed cells (1.51 ± 0.32-fold increase).

In essence, exposure to STX3451 appeared to cause a substantial increase in ROS 48 h after exposure, particularly in MCF-7 cells. Thus, although the radiation did contribute to the increased ROS, the majority of the increase may be attributed to pre-treatment with the compound.

### 2.8. STX3451 Suppressed ATM Expression as Part of the DNA Damage Response

To investigate the effects of STX3451 exposure on DNA damage response signaling (DDR), temporal ATM expression was determined using Western blot analyses 2 and 24 h post-radiation (Figure 9).

Both cell lines exposed to STX3451 displayed suppressed levels of ATM in 2 and 24 h post-radiation. At 2 h, ATM expression increased in MCF-7 cells receiving radiation treatment (14.68 ± 4.08-fold) and combination treatment (8.74 ± 0.08-fold). After 24 h, significant decreases in ATM expression were observed in the radiation-exposed samples (2.54 ± 0.56-fold), the chemo-radiation samples (2.04 ± 0.34-fold) and the STX3451 control (0.74 ± 0.05-fold). Although increased, ATM expression was attenuated in the combination samples compared to the radiation controls at 24 h.

In MDA-MB-231 cells, ATM expression was significantly increased in the radiation treatment at 2- (4.54 ± 1.18-fold increase) and 24-h (1.87 ± 0.48-fold increase). ATM expression was also depressed in STX3451-treated cells (0.37 ± 0.46-fold decrease at 2 h and 0.63 ± 0.26-fold decrease at 24 h). ATM expression was significantly lower in the combination treatment than in radiation alone in the 24 h timeline.

Results indicate that STX3451 suppressed the expression of ATM, a response demonstrated in the radiation controls. The hypothesis generated was that STX3451-induced disruption of microtubule dynamics results in the impaired shuttling of DDR proteins such as ATM. This inability of the cell to shuttle ATM to the site of dsDNA breaks interferes with damage response signaling and leaves the cell more vulnerable to the DNA damaging effects of radiation [39]. Future studies should aim to look at the temporal correlation between the DNA damage (which increased at 48 h) and the decreased ATM expression at the same timepoint. Methods such as fluorescent GFP-tagged ATM and immunocytochemistry would aid in establishing the sub-cellular location of ATM in the various treatment conditions, together with Western blot analysis of protein activation up- and downstream from ATM.

## 3. Discussion

Radiation therapy constitutes an important role in the control of local tumors and certain metastases in many cancer management plans. High doses of radiation have a range of unwanted effects which decrease the patient’s quality of life. Additionally, radioresistance may determine a poor prognosis. Radiosensitizing agents aim to facilitate a better clinical outcome by using lower doses of radiation with sub-therapeutic doses of the drug resulting in an augmented anticancer effect and thereby diminishing the side effects of each modality. Thus, investigating new cost-effective molecules that could serve as efficient radiosensitizers, with minimal toxicity on normal cells, circumventing radioresistance and improving treatment goals, remains topical and relevant. Elucidation of the intracellular pathways induced is an important end goal in itself, as this information may identify new targets which can be selectively induced/inhibited in future novel combinations.

STX3451 has been previously shown to be cytotoxic to cell lines of the rare neurofibromatosis NF1 [40] and NF2 [41] tumors. Radiation therapy can sometimes be used to shrink neurofibromas, but it is often not a clear-cut choice of the preferential treatment modality. If the synergy demonstrated here in these *in vitro* findings can be translated into *in vivo* models, the approach may find applications in such often challenging therapeutic situations. This proof-of-concept study aimed to investigate STX3451 as a potential candidate to pre-sensitize breast cancer cells to radiation in order to lower the required doses and minimize adverse effects, in addition to overcoming radioresistance [25,42].

STX3451 displayed similar cytotoxicity at nanomolar concentrations in MCF-7 and MDA-MB-231 cells (0.074 μM and 0.065 μM, respectively). Enhanced cytotoxicity was observed when the same dose–response curve was exposed to 6 Gy radiation. This was particularly evident at lower doses of the drug. To elucidate intracellular responses behind this observed radiosensitivity, a range of experiments were conducted in which cells were pre-treated with 0.07 µM STX3451 for 24 h prior to 6 Gy radiation. Temporal relationships were established by terminating the experiment 2, 24 or 48 h thereafter and comparing the effect to the radiation and drug experimental controls.

Previous studies reported that STX3451 exposure abrogated microtubule structure and function in MDA-MB-231 cells, resulting in a metaphase block [21,43]. STX3451 was successfully co-crystallized with the αβ-tubulin heterodimer, showing its binding to the colchicine site in atomic detail and its sulfamate group interacting with residues beyond the reach of colchicine itself [35]. The G_2_–M block caused by 2ME2 leads to subsequent apoptotic cell death [42]. Even at sub-lethal doses, 2ME2 analogs disrupted microtubule dynamics and caused disorganized intracellular trafficking of proteins, as seen with disrupted autophagic responses [29].

Similarly, in this study, STX3451 treatment resulted in a metaphase block and induction of apoptosis, as did the cells exposed to radiation. The cells treated with the combination showed a decrease in viability and an increase in metaphase and apoptosis, similar to the drug control group at both 24 and 48 h after radiation. The effects were significantly more pronounced than those produced by the radiation control in MCF-7 cells.

The metaphase block induced by MDAs may lead to mitotic catastrophe and the release of death receptor 3 (DR3), which leads to the activation of mitosis-linked apoptosis [44]. Steroidal sulfamoylated 2ME2 analogs induced apoptosis via the intrinsic and extrinsic pathways, which could be quantified by flow cytometric analysis of labeled Annexin V [29,32,45,46]. Similar assays were conducted in this study. At 48 h post-radiation, STX3451 treatment decreased cell viability and induced apoptosis, whereas the radiation samples only displayed a small reduction in viability and a mild to no increase in apoptosis in both cell lines. The combination experimental samples displayed a significant decrease in viability concurrent with a sharp increase in apoptotic cells, similar to the drug control samples. It may tentatively be deduced that these relatively early effects are more related to the drug exposure at this time. Light microscopy revealed an increase in the number of rounded cells and a reduction in cell density in response to STX3451 and the combination protocol. This correlated with the results of the cell cycle and apoptosis induction analysis.

MDAs may also cause cell cycle arrest in G_1_, with upregulation of cyclin-dependent kinase (CDK) inhibitors [47,48]. An increase in the CDK inhibitors p27 and p15 has been described in MDA-MB-231 and HeLa cells exposed to two structurally related steroidal sulfamoylated 2ME2 molecules [29]. Data from this study revealed a significant decrease in the S-phase at 48 h in the MDCF-7 cells exposed to the compound and the combination regimen, which may point to such a cell cycle block. The G_1_–S checkpoint will halt cell cycle progression at G_1,_ particularly when DNA damage is detected [49]. Cell cycle arrest at this checkpoint is often p53-dependent. ATM is recruited to damaged DNA and p21 is transcribed to arrest the cell cycle via p21-regulated inhibition of cyclins [50].

Exposure to STX3451 caused a significant increase in ROS formation in both cell lines. Specifically, MCF-7 cells displayed significantly increased levels of ROS formation between the 24 and 48 h termination, indicating a later effect from the exposure. The combination sample showed the same pattern but had statistically increased ROS compared to the drug at 24 h and the radiation control at 48 h, perhaps disrupting the temporal sequence of the signaling. THIQs have been found to undergo spontaneous oxidation with concomitant ROS formation *in vivo* [51]. ROS can cause DNA damage such as dsDNA breaks that lead to Mn formation, which may explain the slight genotoxicity seen in compound-exposed cells [40,52]. The chemo-radiation caused an increased ROS production at both timepoints, in both cell lines, statistically more so than in the radiation controls.

To quantify the DNA damage induced by radiation, and to investigate the effect of pre-exposure with the compound on this process, Mn studies were done. The combination treatment resulted in an increased number of cells with DNA damage as well as a greater extent of DNA damage per cell. Radiation after STX3451 exposure demonstrated an augmented effect in Mn formation, with the exception of MCF-7 cells at the 24 h termination time point. The increase in DNA damage in these cells could be ascribed to a number of factors: an increase in the percentage of cells in the most radiosensitive phase of the cell cycle (G_2_–M), disrupted repair protein shuttling from the cytosol to the sites of DNA breaks, increased ROS production induced by the two treatments or a combination thereof.

Markowitz et al. (2016) showed that MDAs disrupt transportation of the DDR element ATM, leading to radio-sensitization of treated cells [41]. ATM plays a pivotal role in the DDR by relaying repair signaling between DNA damage sensors and repair effectors which regulate DNA repair and cell cycle progression. ATM is a protein kinase that is phosphorylated in response to dsDNA breaks and subsequently activates a wide array of effectors to facilitate DNA repair [53,54]. Elevated levels of ATM expression may facilitate tumor resistance to radiation and chemotherapy [55]. ATM-facilitated activation of AKT and nuclear factor kappa light chain enhancer of activated B cells (NF-KB) leads to increased cell survival and decreased apoptosis and facilitates cell migration and metastasis [56]. Tarek et al. found that increased ATM activation in ovarian epithelial tumors correlated with increased resistance to the DNA damaging agent cisplatin [53]. Riballo et al. demonstrated that Artemis phosphorylation by ATM is essential to the repair of dsDNA breaks induced by radiation [57]. In this study, Western blotting was used to analyze the relative protein expression of ATM at 2 and 24 h after 6 Gy radiation exposure. In contrast to the elevated levels of ROS and Mn, expression of ATM was suppressed in combination- and compound-treated samples. It appears that STX3451 suppresses or attenuates ATM expression, which was demonstrated by the compound-exposed cells expressing significantly less ATM than the vehicle controls, which would then theoretically delay the DNA repair.

ATM is present as an inactive dimer primarily in the cytosol of the cell [58]. When double-strand DNA breaks occur, ATM is shuttled to the site of DNA damage and auto-phosphorylates in the presence of the MRN complex, separating into two active monomers [59,60]. Once ATM has been recruited to dsDNA breaks, it phosphorylates histone (H) 2AX, creating gamma-histone 2A variant X (γ-H2AX) foci along the chromatin around the break. γ-H2AX recruits repair proteins to facilitate the rejoining of the broken DNA strands [38]. Along with γ-H2AX formation, activation of Artemis by ATM is essential for the rejoining of dsDNA breaks [57]. Markowitz et al. showed that microtubules are essential for the shuttling of DNA repair proteins (including ATM) from the cytosol to the nucleus, a process disrupted by MDAs [33,41]. Helena et al. demonstrated decreased levels of the double-strand break marker γ-H2AX 2 h after radiation exposure in MCF-7 and MDA-MB-231 cells that had been pre-treated with the 2ME2 analog ESE-16 [28]. This decrease in γ-H2AX was accompanied by depressed levels of the DDR element Ku70 [61]. The depressed levels of both γ-H2AX and Ku70 were in contrast to increased dsDNA breaks detected in the Mn assays. The authors of that study generated the hypothesis that increased ROS formation accompanied by depressed levels of ATM resulted in an increase in dsDNA breaks, but lower levels of γ-H2AX and Ku70 due to disruption of DDR signaling, leading to a decreased long-term survival of the chemo-radiated cells.

Lastly, the long-term viability of the treated cells was investigated using clonogenic studies. Reversibility of the compound’s effect and sufficient DNA repair may cause a reversible decline in cell viability in the early stages of treatment. This was, however, ruled out, as colony formation was synergistically decreased by STX3451 pre-treatment and radiation.

To conclude, STX3451 exposure prior to 6 Gy radiation demonstrated potential radiosensitizing properties in breast cancer cells *in vitro*. Enhanced cytotoxicity was observed when STX3451 was combined with radiation exposure, which was particularly evident at the lower doses of the drug. This chemo-radiation treatment induced a G_2_–M block, increased ROS formation, amplified DNA damage, decreased expression of the DDR element ATM and synergistically decreased long-term cell viability in MCF-7 and MDA-MB-231 cells. Future studies will aim to elucidate further signaling pathways, which could potentially be manipulated to facilitate radiosensitivity. This would aid the design and investigation of targeted proteins or molecular processes to enhance this application. Furthermore, follow-up studies will be needed to ascertain whether the radiosensitization phenomenon observed in this proof-of-concept study conducted in cancer cell monolayers can be extrapolated to living systems. *In vivo* experiments will be designed to determine the treatment dose, optimal time of administration and duration of drug administration prior to radiation, parameters which will also be determined by the drug’s pharmacokinetics. Additionally, these new data further expand the therapeutic potential of STX3451, as well as this general class of non-steroidal compounds.

## 4. Materials and Methods

### 4.1. Materials

Human breast epithelial adenocarcinoma commercial cell lines (MCF-7and MDA-MB-231 cells) were procured from the American Type Culture Collection (ATCC, Rockville, MD, USA). Dulbecco’s modified Eagle’s medium (DMEM) and heat-inactivated fetal calf serum (FCS) were obtained from Life Technologies, Thermo Scientific (Rockford, IL, USA). Methanol (MeOH) and ethanol (EtOH) were supplied by MERCK (Darmstadt, Germany). Penicillin, streptomycin, fungizone, dimethyl sulfoxide (DMSO), glutaraldehyde, phosphate-buffered saline (PBS), crystal violet, propidium iodide (PI), RNase, paclitaxel, etoposide, actinomycin-D, cytochalasin-B, sodium dodecyl sulfate (SDS), β-glycerophosphate, protease cocktail inhibitors, Triton X-100 and all other chemicals of analytical grade were purchased from Sigma-Aldrich Inc. (St. Louis, MO, USA).

### 4.2. Methods

#### 4.2.1. Cell Culture

MCF-7 and MDA-MB-231 cells were cultured in DMEM supplemented with 10% FCS, penicillin (100 units/mL), streptomycin (150 μg/mL) and fungizone (250 µg/mL) in a humidified Forma Scientific incubator (Thermo Fisher Scientific, Waltham, MA, USA) at 37 °C and 5% CO_2_. Cells were sub-cultured and cryo-preserved according to the ATCC guidelines. Exponentially growing cells were seeded at a density of 5.0 × 10^3^ cells/well in 96-well tissue culture plates, 2.5 × 10^5^ cells/well in 6-well plates and 7.5 × 10^5^ cells per 25 cm^2^ cell culture flask. For the clonogenic studies, 8.0 × 10^4^ cells were seeded per 35 mm petri dish.

#### 4.2.2. Treatment Protocol

Cells were seeded at the appropriate density, incubated for 24 h for attachment and subsequently exposed to 0.07 µM (GI_50_ of STX3451). After a 24 h drug exposure, cells were treated with 6 Gy radiation, a concentration at which less than 80% of the cells underwent apoptosis 24 h after exposure [21,29]. Samples were irradiated using a Siemens Oncor Impression Linear Accelerator (SiemensMedical Solutions, Malvern, PA, USA) at the Department of Radiation Oncology, Steve Biko Academic Hospital (Pretoria, Gauteng, South Africa (SA)). Photon beam energy of 6 MV was used with a direct field and a gantry angle of 180°. Source-to-surface distance (SSD) was 100 cm, and the photon beam electronic equilibrium depth (Dmax) was 1.5 cm. A field size of 10 × 10 cm was used with 4.5 cm tissue-equivalent bolus for dose homogeneity. Termination of experiments followed 2, 24 and/or 48 h post-radiation exposure.

#### 4.2.3. Experimental and Method Controls

Negative controls included cells propagated in complete culture medium only. Cells treated with DMSO (<0.05% *v/v*) served as the vehicle control. Positive controls included actinomycin-D (1 µg/mL) to induce apoptosis, etoposide (100 mM) for micronuclei formation and DNA damage and paclitaxel (1 µM) for the induction of ROS. Cells exposed to radiation alone and STX3451 as single treatment modalities served as experimental controls.

#### 4.2.4. Spectrophotometric Quantification of Crystal Violet Staining

Cells were seeded at a density of 5 × 10^3^ cells/well in 96-well tissue culture plates incubated at 37 °C equilibrated with 5% CO_2_ overnight to allow for attachment. Cells were exposed to a concentration range (5 μM–0.01 μM) of STX3451 for 48 h. In parallel, the same experimental concentration range was exposed to 6 Gy radiation 24 h after drug exposure and terminated 24 h thereafter. SDS-treated cells (100% cell death) served as a positive control, and cells treated with DMSO (0.05%) served as the negative control (100% cell viability). On termination, cells were fixed with 100 μL 1% glutaraldehyde and incubated at room temperature (RT) for 15 min. Cells were stained with 100 μL 0.1% crystal violet (RT for 30 min). Cells were gently washed with water. The plates were left to air-dry overnight, and the crystal violet dye was solubilized using 200 μL 0.2% Triton X-100 in ddH_2_O (RT, 30 min). The solution (100 μL) was transferred into 96-well plates, and the absorbance was read at 570 nm using an EPOCH universal microplate reader (BioTek Instruments Inc, Winooski, VT, USA). GI_50_ values were calculated.

#### 4.2.5. Flow Cytometric Analysis of Cell Cycle Progression

Upon termination, cells and culture media were harvested and centrifuged for 5 min at 300× *g*. The pellet was re-suspended in 200 µL ice-cold PBS containing 0.1% FCS and fixed overnight in 4 mL ice-cold 70% EtOH. Cells were incubated in 1 mL PBS containing 40 µg/mL PI, 100 µg/mL RNase A and 20 µL Triton X-100 (0.1%) at 37 °C and 5% CO_2_ for 45 min in the dark. PI fluorescence (FL3) was measured at 617 nm using a Gallios System flow cytometer (Beckman Coulter, Brea, CA, USA) equipped with an air-cooled laser excited at 493 nm, and data from at least 10,000 cells were analyzed using Kaluza flow cytometry analysis software (Beckman Coulter, Brea, CA, USA).

#### 4.2.6. Annexin V Apoptosis Detection by Flow Cytometry

The annexin V–fluorescein isothiocyanate (FITC) Apoptosis Detection Kit (BioLegend, Inc., San Diego, CA, USA) was used. Upon termination, harvested cells were washed with ice-cold PBS and resuspended in annexin V binding buffer containing 100 µL annexin V FITC and 10 µL PI (15 min in the dark at RT). PBS (400 µL) was added, and samples were analyzed using a Gallios System flow cytometer (Beckman Coulter, Brea, CA, USA) equipped with an air-cooled laser excited at 488 nm. Data were captured at emission maxima of 535 nm from a minimum of 10,000 cells. Actinomycin D served as positive method control for apoptosis. Data were analyzed using Kaluza flow cytometry analysis software (Beckman Coulter, Brea, CA, USA).

#### 4.2.7. Long-Term Survival Analysis Using Clonogenic Studies

Seeded and treated cells were allowed to form colonies for 10 days after radiation. Upon termination, cells were fixed with 100 μL 1% glutaraldehyde (RT, 15 min). The glutaraldehyde was discarded and 100 μL 0.1% crystal violet was added (30 min, RT). The plates were immersed under running water for 15 min and allowed to air-dry. Colony formation was evaluated based upon the area the colonies covered in a 6-well plate as well as the density of colonies using the ColonyArea plug-in for Image J 1.48v (National Institute of Health, (NIH), Bethesda, MD, USA). Images were captured with a camera.

#### 4.2.8. Morphological Analysis Using Polarization-Optical Transmitted Light Differential Interference Contrast Microscopy (PlasDIC)

Treated cells were washed with PBS and photographed using a Zeiss Axiovert 40 light microscope equipped with a Zeiss Axiovert MRm monochrome camera (Zeiss, Oberkochen, Germany). Qualitative analysis was completed from 3 biological repeats with a minimum of 5 representative micrographs taken per sample. Actinomycin D served as positive apoptosis control. Micrographs were labeled and processed using Image J (NIH, Bethesda, MD, USA).

#### 4.2.9. Quantification of DNA Damage via Assessment of Micronuclei Formation

Cells were seeded and exposed to the drug and radiation as per protocol, with a modification. Cells were exposed to 12 µM cytochalasin-B 2 or 24 h post-radiation and were incubated further according to their doubling time (29 h for MCF-7 and 38 h for MDA-MB-231). Harvested cells were incubated in hypotonic potassium chloride (KCI) solution (0.14 M) at 37 °C for 5 min. Fixative 1 (13 parts 0.9% NaCl, 12 parts methanol, 3 parts acetic acid) was added to the cell pellet and incubated at RT for 5 min. Fixative 2 (4 parts methanol, 1 part acetic acid) was added to the cell pellet (5 min; RT). All solutions were added drop by drop with continuous agitation. Two drops of the washed resuspended cellular pellet were applied onto degreased microscope slides which were allowed to dry. Slides were stained with 10% Giemsa for 20 min and rinsed thrice with ddH_2_O. A Nikon Optiphot transmitted light microscope (Tokyo, Japan) was used to view the slides, in which 500 binucleated cells from each slide were blind scored for micronuclei. The number of cells with more than one nucleus (replication index) and the number of micronuclei in 500 binucleated cells (dispersion index) were documented. Cells treated with etoposide (1 µM) served as a positive method control.

#### 4.2.10. Quantification of Reactive Oxygen Species (ROS) Using Flow Cytometry

Hydroethidine (HE) was added to harvested cells (7.5 × 10^5^ cells per sample) (final concentration of 20 μM) and incubated for 15 min at 37 °C. Quantification of superoxide production was performed using a Gallios System flow cytometer with an air-cooled laser excited to 488 nm (Beckman Coulter, Brea, CA, USA). Kaluza flow cytometry analysis software version 2.1 (Beckman Coulter, Brea, CA, USA) was used to analyze the data. Histograms were generated by plotting HE emission at 605 nm (FL3 log) on the *x*-axis against cell count on the *y*-axis. Statistical analysis was completed from 3 biological repeats, with a minimum of 20,000 cells per run. Actinomycin D served as positive method control.

#### 4.2.11. Analysis of ATM Expression as Part of the DNA Damage Response Signaling Using Western Blot

Upon termination, cells were washed with ice-cold PBS. Ice-cold radioimmunoprecipitation assay buffer (RIPA) (150 mM NaCl, 10 mM Tris(hydroxymethyl)aminomethane-hydrochloric acid (Tris-HCl), pH 7.4, 0.1% SDS, 0.5% sodium deoxycholate, 1 mM ethylenediaminetetraacetic acid (EDTA), 1 mM ethylene glycol tetra-acetic acid (EGTA), protease inhibitor cocktail 2 and phosphatase cocktail) was added to the cells (200 μL; on ice; 5 min). Lysed cells were collected by scraping and centrifuged at 10,000 rpm (30 min; 4 °C) to collect the supernatant (kept on ice). The Pierce BCA protein assay kit (Thermo Fisher Scientific, Waltham, MA, USA) was used to determine the protein concentration of each sample. Cytosolic extract (20 μL) with 100 μL of the pre-mixed reagents A and B (50:1) was incubated for 30 min at 37 °C, and the absorbance was read at 570 nm using an ELx800 Universal Microplate Reader (Bio-Tek Instruments Inc., Winooski, VT, USA). Protein concentrations were calculated using a bovine serum albumin (BSA) standard curve (0.1 mg/mL–0.4 mg/mL). Cellular protein (25 μg per sample) was loaded into each NuPAGE gel well together with 4× NuPAGE loading buffer containing 2.5% β-mercaptoethanol, after denaturation at 96 °C for 5 min. Proteins were resolved via electrophoresis on a NuPAGE 4–12% Bis-Tris Gel at 120 V for 90 min in a 1× 3-morpholinopropane-1-sulfonic acid (MOPS) migration buffer (20× 50 mM MOPS, 50 mM Tris, 1 mM EDTA, 0.1% SDS, pH 7.7). Separated proteins were transferred to a 100% MeOH activated polyvinylidene fluoride (PVDF) 0.2 μm membrane (Amersham Hybond, GE Healthcare Life Sciences, Marlborough, MA, USA) with a transfer buffer containing 48 mM Tris, 39 mM glycine, 20% MeOH and 0.0375% SDS (4 °C; 120 V; 2 h). Membranes were incubated in 2% BSA in 0.2% PBS-Tween for 20 min. Membranes were incubated for 16 h (4 °C) in the primary antibody cocktail containing 1:1000 monoclonal anti-ATM antibody (Abcam, Cambridge, UK) in 2% BSA in 0.2% PBS-Tween. Membranes were washed 3 times with 0.2% PBS-Tween and incubated for 1 h at RT with the secondary antibody (anti-mouse IgG (H + L) antibody raised in goat labeled with horseradish peroxidase (Abcam, Cambridge, UK) at 1:10,000 in 2.5% milk). Membranes were washed three times with 0.2% PBS-Tween. Proteins were visualized using a ChemiDoc MP (Bio-rad, Inc., Hercules, CA, USA) after activating horseradish peroxidase activity with Pierce ECL Western blotting reagent (Thermo Fisher Scientific, Waltham, MA, USA). Membranes were standardized using monoclonal anti-glyceraldehyde-3-phosphate dehydrogenase (GAPDH) antibody produced in mouse (1:5000) (Abcam, Cambridge, UK). Band size was determined using densitometric data of band intensities by using Image Lab version 5.2.1 (Bio-Rad Laboratories, Inc., Hercules, CA, USA). Quantification followed standardization using band intensities obtained from GAPDH as loading control. Three biological repeats were analyzed for both cell lines.

### 4.3. Statistical Analysis

Qualitative data were obtained from light microscopy performed in triplicate. Semi-quantitative (micronuclei (Mn) and Western blotting) and quantitative data (spectrophotometry and flow cytometry) were obtained from a minimum of three independent biological repeats (n ≥ 3). Flow cytometric data from a minimum of 10,000 cells were analyzed using Kaluza Analysis Software, version 2.0 (Beckman Coulter, Brea, CA, USA). Western blot analysis was performed on Image Lab, version 6.0 (Bio-Rad Laboratories, Inc., Hercules, CA, USA). Statistical analyses employed the analysis of variance (ANOVA) single-factor model and two-tailed Student’s *t*-test. *p*-values < 0.05 were considered as statistically significant.

## Figures and Tables

**Figure 1 molecules-27-03819-f001:**
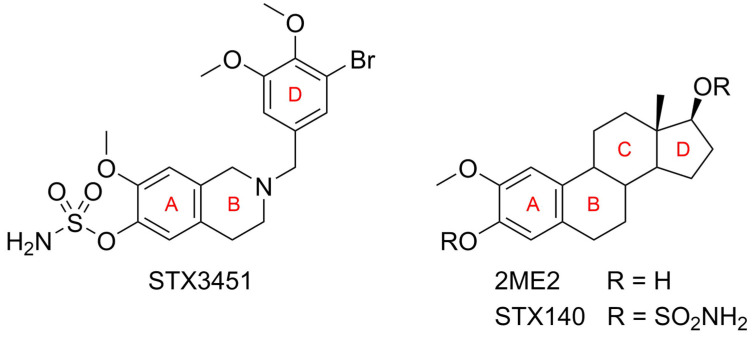
Structures of 2-(3-bromo-4,5-dimethoxybenzyl)-7-methoxy-6-sulfamoyloxy-1,2,3,4-tetrahydroisoquinoline (STX3451), 2-methoxyestradiol (2ME2) and 2-methoxyestradiol-3,17-*O*,*O*-bis-sulfamate (STX140), showing the ring mimicry.

**Figure 2 molecules-27-03819-f002:**
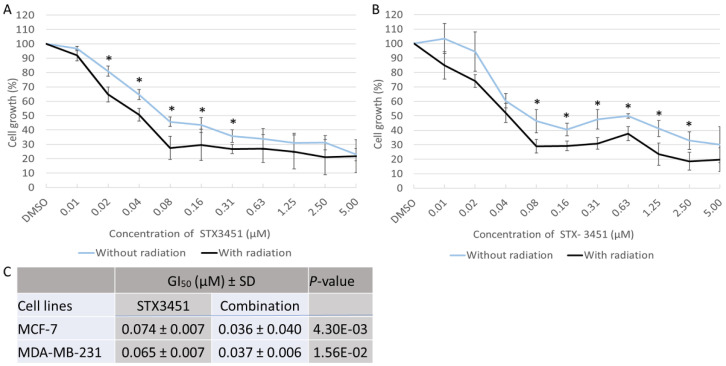
Dose–response curve of STX3451 either without radiation (light-grey line) or with radiation (dark-grey line) used in determination of the concentration required to inhibit growth of MCF-7 (**A**) and MDA-MB-231 (**B**) cells by 50% (GI_50_). Cell growth was determined relative to DMSO. Data points show the mean value from a minimum of 3 repeats, each with an n ≥ 3. Error bars indicate standard deviation (SD), * indicates statistical significance (*p*-value < 0.05) (**C**).

**Figure 3 molecules-27-03819-f003:**
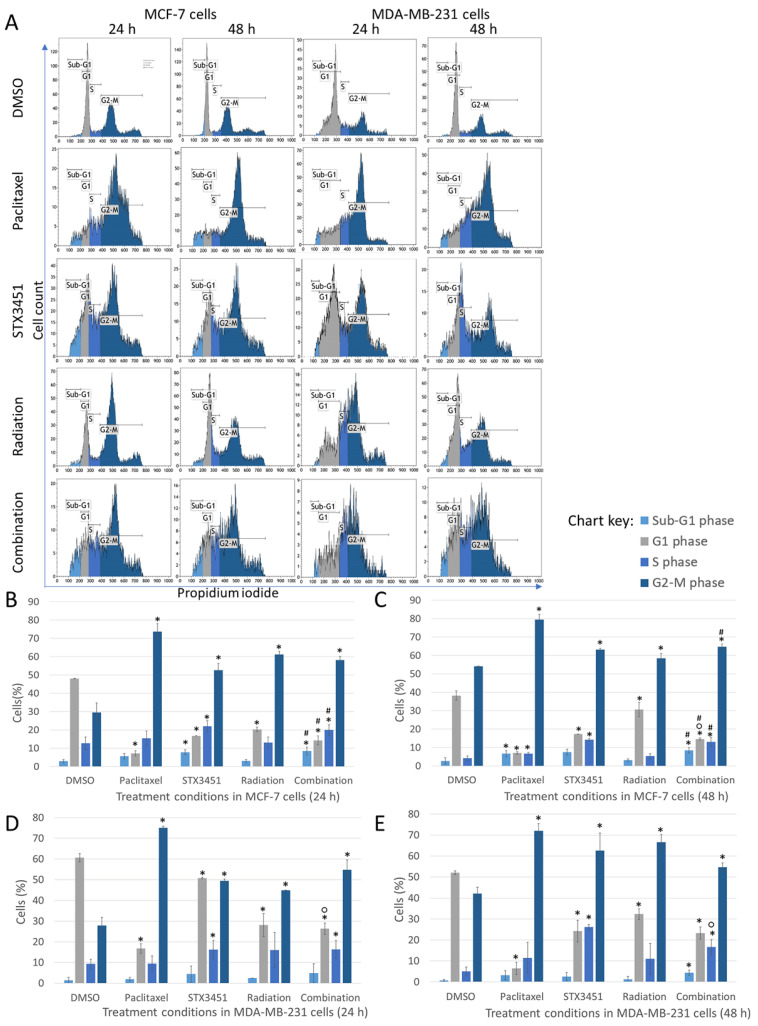
Analysis of the cell cycle progression in MCF-7 and MDA-MB-231 cells terminated 24 and 48 h after exposure to radiation. Histograms (**A**) depict cell count plotted against PI fluorescence (FL3). Bar charts depict the mean values for MCF-7 cells exposed for 24 h (**B**) and 48 h (**C**), as well as MDA-MB-231 cells exposed for 24 h (**D**) and 48 h (**E**) gathered from 3 biological repeats. Error bars indicate the standard deviation. Statistical significance (*p*-value < 0.05) when compared to DMSO is represented by *, ° when compared to compound-exposed cells and # when compared to radiation-exposed cells.

**Figure 4 molecules-27-03819-f004:**
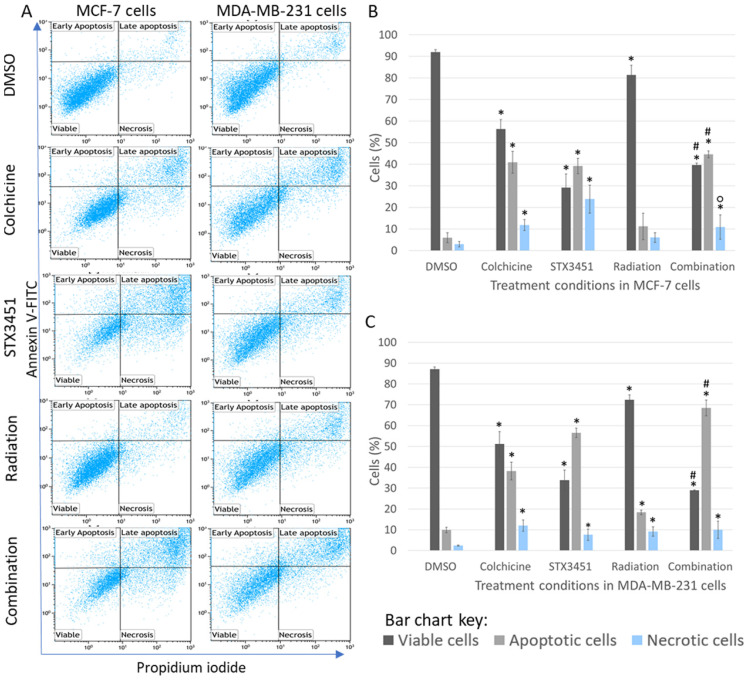
Apoptosis studies. (**A**) Flow cytometric dot plots of annexin V–FITC (FL1) plotted against PI (FL3) in MCF-7 and MDA-MB-231 cells terminated 48 h after radiation exposure. DMSO was used as the vehicle control. At this early stage after treatment, apoptotic effects were observed in the STX3451/6 Gy radiation combination treatment in keeping with the drug-treated cells, but more so than in the radiation-only exposed cells. Bar charts of MCF-7 cells (**B**) and MDA-BM-231 cells (**C**) represent the mean of data collected from 3 biological repeats. Error bars indicate standard deviation. * represents statistical significance (*p*-value < 0.05) when compared to DMSO-exposed cells, # when compared to radiation-exposed cells and ° when compared to compound-exposed samples.

**Figure 5 molecules-27-03819-f005:**
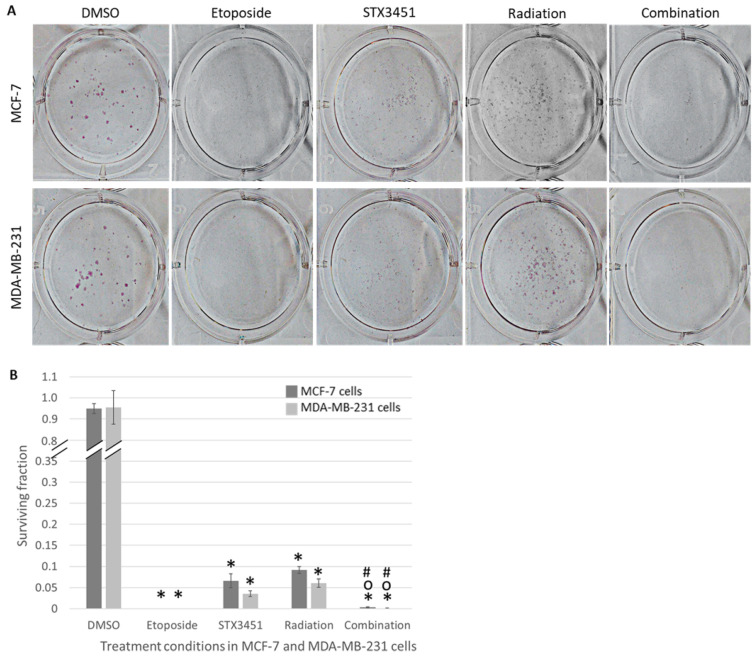
Colony formation in MCF-7 and MDA-MB-231 cells exposed to control and treatment conditions (**A**). Colony formation was allowed to progress for 14 days after cells had been exposed to DMSO, etoposide, STX3451, 6 Gy radiation and combination treatment. Cellular seeding densities were adjusted according to treatment conditions, and surviving fraction values of colonies were adjusted according to the number of cells seeded for each treatment. Bar graph (**B**) represents data gathered from 3 biological repeats. A decrease in colony formation was observed in all treatment conditions, but more so in the combination treatment samples. Error bars represent standard deviation * Statistical significance (*p*-value < 0.05) when compared to DMSO-exposed cells, ° compared to compound-exposed cells and # compared to radiation-exposed cells.

**Figure 6 molecules-27-03819-f006:**
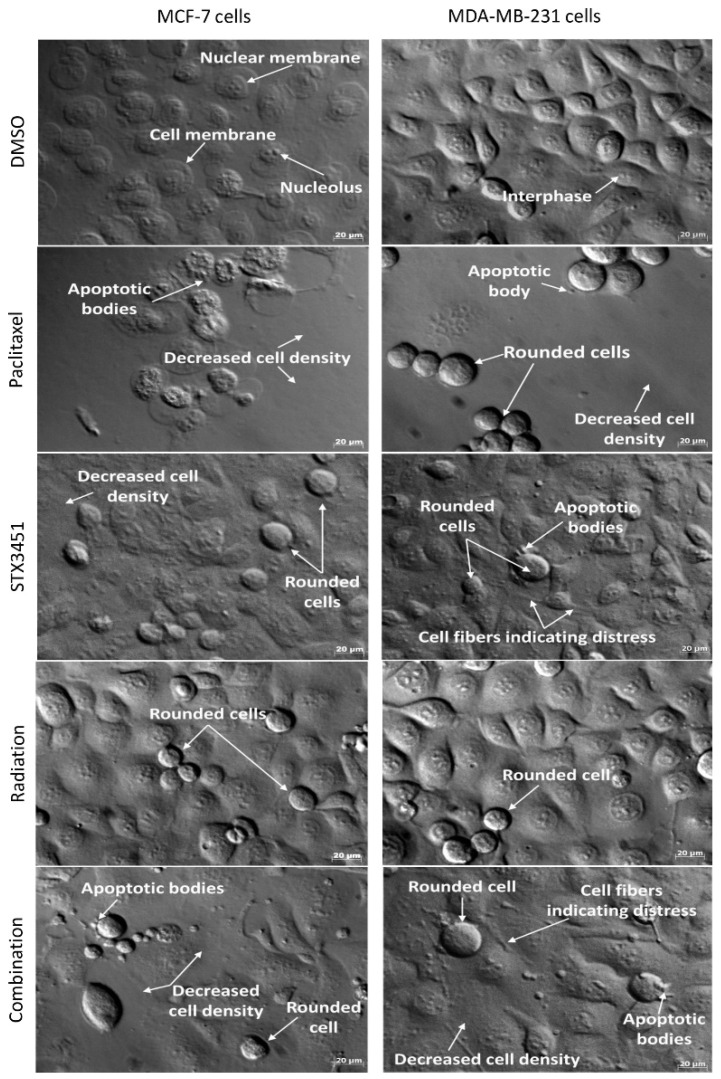
PlasDIC micrographs depicting MCF-7 and MDA-MB-231 cells after exposure to STX3451 and radiation. Cells exposed to DMSO displayed confluent cells with no signs of morphological change. Cells exposed to paclitaxel served as positive apoptosis control. Cells exposed to STX3451 and 6 Gy radiation both showed cell rounding. Cells exposed to 0.07 µM STX3451 prior to radiation displayed a decrease in cell density, as well as cell rounding and apoptotic bodies. Micrographs were taken at 40× magnification, with 3 biological repeats. The scale bar = 20 µm.

**Figure 7 molecules-27-03819-f007:**
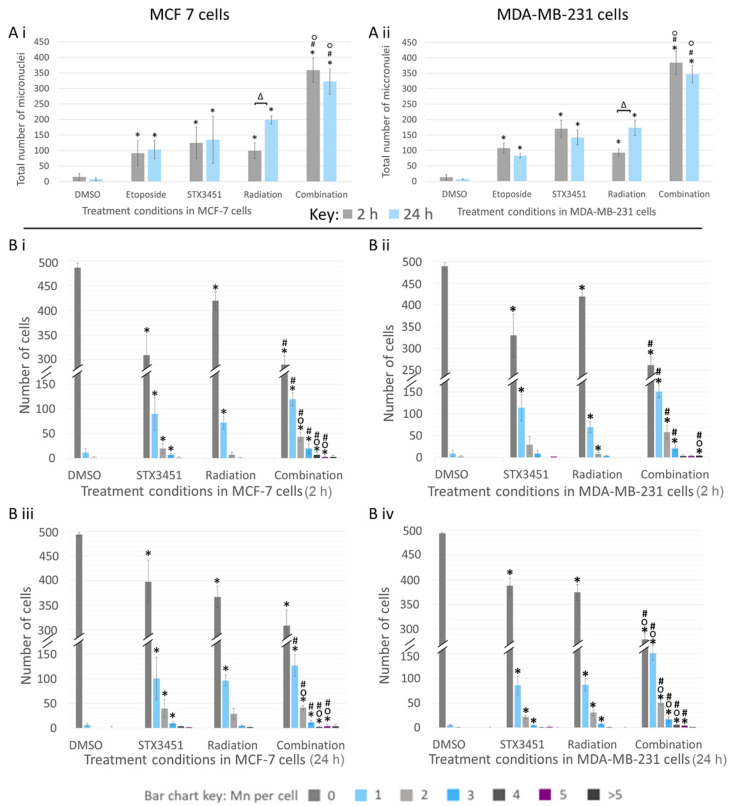
Micronuclei analyses in treated MCF-7 and MDA-MB-231 cells terminated 2 and 24 h after radiation. Cells were exposed to 0.07 µM STX3451 prior to 6 Gy radiation. (**A**) Total number of Mn per sample is displayed in MCF-7 (**Ai**) and MDA-MB-231 cells (**Aii**). (**B**) The number of Mn per MCF-7 cell 2 h (**Bi**) and 24 h (**Biii**) after radiation, and the number of Mn per MDA-MB-231 cell at 2 h (**Bii**) and 24 h (**Biv**) after radiation. The chemoradiated cells displayed a greater number of total Mn per sample, as well as showing an increased number of cells with more extensive DNA damage. Bar charts represent the mean of 3 biological repeats. Error bars indicate SD. Statistical significance is indicated by * when compared to DMSO, ° compared to the STX3451 control and # compared to radiation control (*p*-value < 0.05). Δ indicates a statistical difference between timelines.

**Figure 8 molecules-27-03819-f008:**
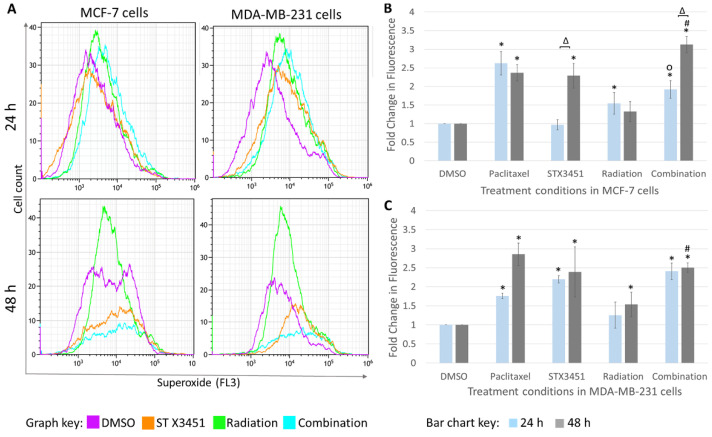
Flow cytometric overlay histograms (**A**) of reactive oxygen species analysis of MCF-7 and MDA-MB-231 cells terminated 24 and 48 h after radiation. STX3451 and combination treatment show a higher mean value after 48 h. Bar chart represents the mean gathered from 3 biological repeats in (**B**) MCF-7 cells and (**C**) MDA-MB-231 cells. * represents statistical significance (*p*-value < 0.05) when compared to DMSO-exposed cells, # when compared to radiation only and ° compared to STX3451. Δ indicates statistical significance between the 24 and 48 h termination times.

**Figure 9 molecules-27-03819-f009:**
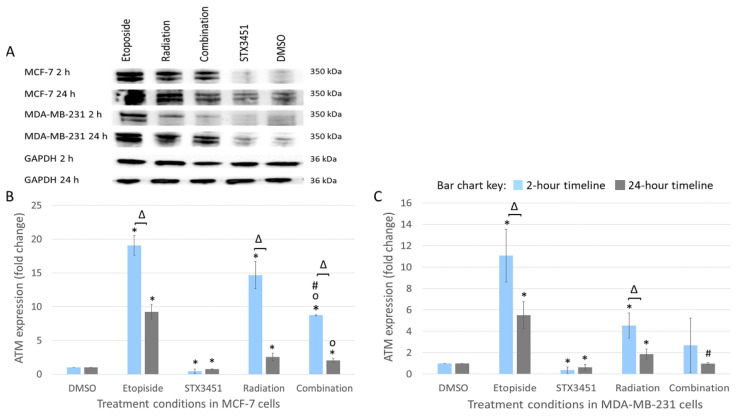
Depiction of the Western blot protein bands (**A**) showing ATM expression 2 and 24 h after radiation. ATM (molecular weight (MW) 350 kDa) levels decreased in both cell lines in the combination-treated cells relative to the radiation treatment. GAPDH (36 kDa MW) served as a housekeeping protein. Bar graphs for MCF-7 cells (**B**) and MDA-MB-231 cells (**C**) showing mean values of 3 biological repeats, with error bars indicating SD. Statistical significance (*p*-value < 0.05) is indicated by * when compared to DMSO, ^o^ compared to STX3451 and # compared to radiation treatment. Δ indicates statistical significance between the 2 and 24 h termination points.

## Data Availability

Appendix A are provided as Appendix A.

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
