# Peer review of "Cell Fate following Irradiation of MDA-MB-231 and MCF-7 Breast Cancer Cells Pre-Exposed to the Tetrahydroisoquinoline Sulfamate Microtubule Disruptor STX3451"

_molecules, 2022, doi:10.3390/molecules27123819_

Round 1
Reviewer 1 Report
Excellent research paper conducting well planned studies. I would like to congratulate the authors for their nice study.
I am wondering what would be the effect of same irradiation on non-cancer cells. Please include if you have any related data.
Reviewer 2 Report
The paper entitled Cell fate following irradiation of MDA-MB-231- and MCF-7
breast cancer cells pre-exposed to a tetrahydroisoquinoline sulfamate microtubule disruptor proposed to text the effect of STX3451-exposure prior to 6 Gy radiation. The study was well-designed, has a good quality and scientific merit. I just recommend the authors to review the key words as if they are listed in the scientific descriptors of PUBMED to make it easier accessible by oher authors.
Reviewer 3 Report
The paper investigates the effect of a pretreatment with the compound STX3451 of two breast cancer cell lines prior to radiation treatment and show increased apotosis upon the combined treatment.
The graphics are poor quality, very hard to distinguish between the different lines (i.e Figures 2, 3,8) when printed in black and white. The frames need to be removed and the annotation needs to be improved. The figures are not intuitively presented. Figure 7 should be completely reorganized to make it clearer and easier to read.
Figure 4 has distorted writing in the Bar chart key.
Many Figures could be better described in the text, for examples Figures 2 and 3. Many points shown in the Figure are not highlighted in the text.
GI50 is not clearly defined.
The authors could improve in highlighting the bigger impact and knowledge gap their study fills.
There are typographic and grammatical errors (i.e page 4 "AT should not be all capitalized, and "...was observed with the..." and spaces missing.
In Figure 1, it would be helpful to show the A,B,D rings that are mentioned in the text. Maybe it would also be good to show the parent steroid.
Why did the authors not display Figure 2 in a logarithmic scale?
Almost all numerical numbers need to be corrected for significant figures.
In Figure 4 the authors show Colchicine controls but never introduce or discuss the control in the section.
On page 7, the authors introduce briefly a new hypothesis but don't state how it can be tested. Further discussion would be helpful.
On Page 7 the authors suggest further studies without any detailed suggestion what these studies would consist of and what questions they would help answering.
On Page 9, last paragraph, what would be the therapeutical relevance on a 24h pretreatment? Is this achievable?
Section 2.7 has a good conclusion sentence at the end to highlight the section, all sections could benefit from that.
The language and writing style differs largely between section, making some sections easier to read and understand than other.
In the discussion many important aspects of the therapy are mentioned that could be moved to the introduction.
What is the toxicity of STX3451 in normal cells?
The authors suggest a synergistic effect of the combined treatment. Could this be proven empirically? Maybe the application of TU (toxic units; TU=TUradiation/TUSTX or something similar) might be beneficiary.
Page 14, 3rd paragraph is a listing of different papers investigating ATM. It would be beneficiary to compare in more detail and put the hypothesis more into context.
In the Method section 4.2.2 needs more detail regarding cell density. Cell density is important when treatment with chemicals since it can influence rate of exposure and metabolism.
Method 4.2.5 "were analysed with Data were analysed" needs to be grammatically corrected.
Method 4.2.9 KCl. A protocol is mentioned but not referenced
4.2.10. cells are harvest but it is not clear how many/cell density.
4.2.11 what is the MOPS concentration of the running buffer?
Reviewer 4 Report
This manuscript investigated the combination effects of STX345 and radiation treatment in breast cancer cell lines. Although there was no animal experimental study in vivo, this manuscript was well prepared. The results are significant and discussions are reasonable. Figure 9A, the word "compoud" above protein lane should be changed into "STX345“. If the authors can provide GAPDH protein lane at 2 h and 24 h simultaneously, that will be better than present data.
